# The Protective Effects of Silkworm (*Bombyx mori*) Pupae Peptides on UV-Induced Skin Photoaging in Mice

**DOI:** 10.3390/foods13131971

**Published:** 2024-06-21

**Authors:** Xiao Lin, Yuting Fan, Liuying Li, Jiamin Chen, Songyuan Huang, Wenqi Yue, Xuli Wu

**Affiliations:** 1Medical School of Pharmaceutical Sciences, Health Science Center, Shenzhen University, Shenzhen 518060, China; 2070245003@email.szu.edu.cn (X.L.); 2110245009@email.szu.edu.cn (L.L.); 2210245053@email.szu.edu.cn (S.H.); 2School of Public Health, Health Science Center, Shenzhen University, Shenzhen 518060, China; fanyuting@szu.edu.cn (Y.F.); 2100243004@email.szu.edu.cn (J.C.); 2200243010@email.szu.edu.cn (W.Y.)

**Keywords:** silkworm pupae, hydrolyze, peptides, anti-photoaging, UV radiation

## Abstract

Silkworm (*Bombyx mori*) pupae are popular edible insects with high nutritional and therapeutic value. Currently, there is growing interest in the comprehensive application of silkworm pupae. In this study, peptides that exhibited anti-photoaging activity were obtained from silkworm pupae protein, aiming to investigate the protective effects and potential mechanisms of silkworm pupae peptides (SPPs) on skin photoaging. The results showed that SPPs were composed of 900 short peptides and could effectively alleviate skin photoaging progression. They significantly eliminated excessive production of ROS and MDA; meanwhile, they also renovated the antioxidant enzyme activities. The biomarkers related to collagen synthesis and degradation, including hydroxyproline, interstitial collagenase, and gelatinase, demonstrated that SPPs could suppress collagen degradation. Histopathological results showed that SPPs could reduce the inflammatory infiltrate and the thickness of the dermis and epidermis, as well as increase the collagen bundles and muscle fibers. The histopathological and biochemical results confirmed that SPPs could alleviate photoaging by inhibiting abnormal skin changes, reducing oxidative stress, and immune suppression. Overall, these data prove the protective effects of SPPs against the photoaging process, suggesting their potential as an active ingredient in skin photoaging prevention and therapy.

## 1. Introduction

Silkworm (*Bombyx mori*) pupae are popular edible insects. In East Asia, silkworm pupae are historically used as edible insects because of their high nutritional value [1]. Especially in China, silkworm pupae have been consumed as a sustainable food source for more than 2000 years. However, for a long period, the majority of silkworm pupae have been regarded as by-products of the sericulture industry and mainly used as fertilizer despite their high nutritional and therapeutic value [2]; thus, this has led to profound economic and resource waste.

With the development of the food industry, many studies on edible insects have been carried out globally during the past 2 decades, including nutrient ingredient identification [3], nutritional value assessment [4], and further processing [5]. Numerous studies have confirmed that silkworm pupae possess excellent nutritional qualities and contain abundant nutrients, such as proteins, lipids, and vitamins; the protein includes a complete set of 18 amino acids, among which the essential amino acids account for over 55% [6]. In addition to macronutrients, silkworm pupae also contain multiple inorganic substances [7]. Though silkworm pupae have remarkable nutritional qualities, the allergenicity limits their application in the food industry. With the deepening understanding of silkworm pupae, the protein hydrolysate and peptides (SPPs) have gained more attention due to their excellent stability, high bioaccessibility, and low allergenicity [8]. What is more, researchers have found that hydrolysis of protein would generate multiple biological functions, such as enhancing immunity [9] and anti-tumor [10] and antioxidant properties [11]. Thus, a wide range of enzymes have been used during hydrolysis to optimize the generation of hydrolysates with different functionalities while reducing the allergenicity. It was confirmed that two short peptides from the silkworm pupae protein displayed a blood-sugar-modulating function through a mixed-type DPP-IV inhibition mode, which was validated by in vitro analysis [12]. Peptides catalyzed by a series of enzymes exhibited inhibitory activity on the angiotensin I-converting enzyme, which is associated with regulating blood pressure [13]. SPPs produced by protease and alcalase could suppress the differentiation of preadipocytes and adipogenesis by modulating signal transduction pathways [14]. However, the release of anti-photoaging peptides from silkworm pupae protein has not been reported yet.

Photoaging is a common type of skin aging caused by solar, particularly prolonged UV radiation, and is mainly characterized by histopathologic and clinical degenerative changes, leading to epidermis hyperplasia, loose skin, and coarse wrinkling [13]. UV radiation can not only induce photoaging but also generate ROS and proinflammatory cytokines, and promote MMP expression [15]. Multiple defenses against UV damage, including physical and chemical protections [16], and eating a combination of nutrient-dense foods can help avoid skin photoaging. In terms of nutrients, there is increasing evidence implying that dietary supplementation, for instance, proteins, peptides, vitamins, and functional polysaccharides [17], plays an important role in ameliorating free radicals and oxidative stress triggered by UV light. As is well known, L-ascorbic acid possesses the predominant role in the chemical and physical protection of skin photoaging; however, some novel peptides were gradually found to have similar functionalities with L-ascorbic acid. For example, clinical trials showed that oral administration of collagen peptides increased the skin collagen content and improved facial moisture [18], and some bioactive peptides from marine animals exhibited strong light protection and light repair properties [19]. The most well-studied silkworm sericin protein was proven as a potent antioxidant and photoprotective agent against UV radiation [20]; meanwhile, it could prevent mitochondrial damage and inhibit melanogenesis [21]. Silkworm bioactive peptides may exhibit potent anti-photoaging and skin protection properties; however, the investigation into their underlying mechanisms remains at a nascent stage.

SPPs possess potent biological activities; therefore, there is a pressing demand for their further comprehensive utilization to achieve economic and environmental benefits. The objective of this study was to investigate the beneficial effects of SPPs on UV irradiation-induced skin damage. SPPs were prepared and isolated from silkworm pupae protein, and then the anti-photoaging effects were investigated and compared both in vivo and in vitro. The possible mechanisms were explored macroscopically and microscopically. This study might provide technical support for the development of silkworm pupae protein for the food, pharmaceutical, and cosmetic industries.

## 2. Materials and Methods

### 2.1. Materials

Silkworms (P50 strain) for pupae collection were provided by the Guangdong Academy of Agricultural Sciences (Guangzhou, China) and were raised in standardized conditions in our lab. Female Kunming mice (6–8 weeks old, 22–26 g) were obtained from Guangdong Medical Laboratory Animal Center (Foshan, China). Alcalase (200 U/mg) was purchased from Macklin (Shanghai, China). Hematoxylin and eosin (H&E), Masson’s trichrome, and Gomori staining kits were purchased from Solarbio Life Science (Beijing, China). Enzyme-linked immunosorbent assay (ELISA) kits, including ROS (E-BC-K138-F), MDA (A003-1-2), IL-6 (E-EL-M0044c), IL-10 (E-EL-M0046c), SOD (E-BC-K020-M), GSH-Px (E-BC-K096-M), CAT (A003-1-2), MMP-1 (E-EL-M0779c), and MMP-9 (E-EL-M3052), were purchased from Elabscience Biotech (Wuhan, China). All other chemicals were analytical grade and obtained from Macklin.

### 2.2. Extraction and Preparation of SPPs from Silkworm Pupae Protein

Silkworm pupae protein was prepared as described in our previous research with slight modifications [22,23]. Briefly, the silkworm pupae were ground into powder with liquid nitrogen and degreased in ethyl acetate to remove lipids. Then, the sample was dispersed in deionized water and pH was adjusted to 10.0 with NaOH (1.0 mol/L). After centrifugation at 5000 r/min for 20 min at 4 °C, the precipitation was harvested and dissolved in PBS buffer (PH 7.0). Hydrolysis parameters were optimized based on response surface methodology. A Box–Behnken design with 4 variables was used to determine the response pattern and establish a model. The 4 variables were temperature (A), pH (B), enzyme dosage (C), and time (D), with 3 levels of each variable, while the response value was tyrosinase inhibition rate. The symbols and levels are shown in Table 1.

### 2.3. Tyrosinase Inhibitory Activity Assay

First, 40 μL samples and 40 μL tyrosinase (2 mg/mL) were mixed in a well of a 96-well plate and gently shaken for 30 s. After incubating for 5 min at 37 °C, 50 μL of L-tyrosine (0.2 mg/mL) was injected into the system and then reacted for 25 min at the same temperature. The absorbance of the products was recorded at 290 nm in a Synergy HTX Multi-Mode Microplate Reader (BioTek Instruments Inc., Winooski, VT, USA). Each sample was analyzed in triplicate. Tyrosinase inhibition was determined by the remaining activity against the negative control.
Tyrosinase inhibitory activity% = [(A1 − A2) − (A3 − A4)]/(A1 − A2)(1)
where A1 is the absorbance of the sample group with tyrosinase, A2 is the sample group without tyrosinase, A3 is PSB buffer with tyrosinase, and A4 is PBS buffer.

### 2.4. Peptide Sequence Analysis

An online nanoflow liquid chromatography–tandem mass spectrometry performed on an Easy nanoLC 1200 system (ThermoFisher Scientific Inc., Waltham, MA, USA) coupled to a Thermo Scientific LTQ-XL linear ion trap mass spectrometer was used for peptide identification. The peptide sequences were analyzed by PEAKS Studio 8.5 to match the sequence of the theoretical spectrum downloaded from the UniProt peptide retrieval database.

### 2.5. Animal Experiments

Female Kunming mice were housed in an environment with a temperature of 20 ± 2 °C and humidity of 50 ± 5%, under a 12 h light/dark cycle, and given free access to standard diet and water. All experimental procedures were carried out under the guidelines of national standards outlined in “Requirements for Laboratory Animal Facilities and Care” (Approval No. IACUC-202300134). Prior to the start of the experiment, mice were acclimatized for 1 week. Mice were randomly classified into 6 groups (n = 7/group): (1) NC group, UV-unexposed, saline-solution-treated; (2) MC group, UV-exposed, saline-solution-treated; (3) PC group, UV-exposed, L-ascorbic acid-treated at 40 mg/bw.kg/d; (4) SPP groups, UV-exposed, SPP-treated (low dose at 0.25 g/kg/d, middle dose at 0.5 g/kg/d, high dose at 1 g/kg/d). The dorsal area of each mouse was shaved. All mice except for the NC group were irradiated 3 times a week for 8 weeks. The radiation intensities were 850 μW/cm^2^ (UVA, 340 nm) and 592.3 μW/cm^2^ (UVB, 313 nm), respectively. The total dose of UV irradiation was 2.4 J/cm^2^. Each time after irradiation, SPPs were given through intragastric administration. The animals were euthanized after 8 weeks of UV exposure, and biopsies were obtained from the dorsal skin for histological analysis. Thymuses, spleens, and serum were extracted. The thickness of the dorsal area was determined immediately after euthanasia by an electronic vernier scale. The fresh skin was dehydrated, and the skin moisture value was calculated. The remaining skin specimens were stored at −80 °C.

### 2.6. Macroscopic Examination of Mice Dorsal Skins

The dorsal skins of mice were photographed. The macroscopic visual aspects shown in Table 2 were evaluated by visual scores following the grading scale described by Ferrucci [24]. A pinch test was carried out to determine skin elasticity. The midline of the dorsal skin was picked up with fingers to a degree until the feet of the animal lightly touched the table. Skin recovery time was calculated immediately.

### 2.7. Histological Analysis

The dorsal skin samples were fixed with 4% paraformaldehyde, dehydrated, and embedded in paraffin. Sections of 4 μm were prepared and submitted to different staining techniques, which included hematoxylin and eosin (H&E), Masson’s trichrome, and Gomori stain. Microscopic observations were made using a Nikon LV-150 microscope.

### 2.8. ROS, MDA Levels, and Antioxidant Enzyme Activities

Skin tissue specimens were weighted and combined with saline solution at a ratio of 1:9 (*w*/*v*), homogenized on ice, and centrifuged. ROS levels and malondialdehyde (MDA) contents in 10% homogenate supernatant were measured with commercial kits following the manufacturer’s procedures. Antioxidant enzyme activities, which included superoxide dismutase (SOD), Glutathione peroxidase (GSH-Px), and catalase (CAT), were determined using commercial kits.

### 2.9. Analysis of Levels of Inflammatory Markers and MMPs

Effects of photoaging induced by UV irradiation were evaluated by measuring related inflammatory markers (IL-6, IL-10) and MMP levels (MMP-1, MMP-9) in the supernatant were evaluated by ELISA with commercial kits.

### 2.10. Statistical Analysis

All experiments were performed in triplicate, and the results are presented as the mean ± deviation. The results were subjected to a one-way analysis of variance (ANOVA). Statistical analyses were performed and the figures were produced in GraphPad Prism 7.0 (GraphPad Software Inc., San Diego, CA, USA).

## 3. Results and Discussion

### 3.1. Molecular Weight and Peptide Composition of SPPs

The combined effect of temperature, pH, enzyme dosage, and time on tyrosinase inhibition activity is presented in Appendix A. Statistical parameters such as *F*-test probability and *p*-value are also summarized in Appendix A. The results for the *p*-value were below 0.01, indicating that the fitness of the model was statistically significant; meanwhile, the lack-of-fit value was 0.1005, further indicating that the model was significant. As shown in the response surface graph, the optimum hydrolysis conditions for the SPPs, determined to obtain the maximum tyrosinase inhibition activity, were 52 °C, pH 8.0, 5.0 h, and 3200 U/g, respectively. Based on the optimum hydrolysis conditions, the peptide composition of the SPPs was further identified. There were 900 peptides, and the most abundant 10 peptides are listed in Table 3. The molecular weight ranged from 300 Da to 1600 Da, and the overwhelming majority (95.4%) of the peptides in the SPPs were distributed below 1000 Da. It was reported that peptides below 1000 Da exhibited strong antioxidant activities; thus, there is a strong hint about the antioxidant activities of SPP.

### 3.2. Macroscopic Visual Appearance in Photoaging Skin

Long-term UV irradiation could cause exogenous skin aging, which is mainly manifested as typical photoaging symptoms such as dry, loose skin, wrinkles, and thickening. Whether the SPPs have any benefits in reliving irregular erythema and wrinkles was analyzed in this part. The macroscopic appearances of mice after 8-week UV exposure are shown in Figure 1. The NC group and PC group did not exhibit any UV-induced skin damage as expected. The MC group, which was UV-exposed and treated with saline, showed coarse wrinkles; meanwhile, inflammatory infiltrations occurred. As to the SPP groups, topical administration of SPPs inhibited the formation of UV-induced erythema and roughness in a dose-dependent manner. In the SPP-H group, mice skin appeared as healthy and smooth as in the NC and PC groups, whereas, the SPP-L group displayed slight sunburn and erythema. The macroscopic visual appearance suggested that ingestion of a sufficient dose of SPPs might be beneficial in inhibiting UV-induced skin damage.

### 3.3. Visual Scores, Skin Elasticity, and Thickness

The visual scores, pinch test, and skin thickness are demonstrated in Table 4. The visual score for the NC group was set as 0. The score for the MC group was remarkably higher than that of other groups. When treated with SPPs and L-ascorbic acid, the decline in the scores was noteworthy, indicating that SPPs could effectively prevent UV-induced skin damage. Furthermore, the scores of the SPP-treated groups decreased in a dose-dependent way, consistent with the macroscopic appearances. Pinch tests were performed to quantify the skin elasticity. As shown in Table 4, the visual score and recovery time in the MC group were significantly different (*p* < 0.05) from the other groups. The skin recovery time was longer in the SPP-L group than in the SPP-M and SPP-H groups, which means a higher dosage would be more effective. However, the reduction in recovery time with a 1 g/bw.kg/d dosage of SPPs was significant and could not match the PC values. Similar results were found in skin thickness, yet there was no significant difference between groups. Thus, treated with repeated UV-irradiation would obviously increase the epidermal thickness of mice, and SPPs could alleviate the skin-thickening in a dose-dependent way.

### 3.4. Histological Evaluation of Skin Tissue

The pathological process of photoaging from a multitude of epidermal and dermal photodamage is illustrated by different staining methods. As demonstrated in Figure 2, the skin structure of the NC group was intact with a clear outline, and the epidermis was orderly arranged and covered by a thin layer of keratin. In addition, the collagen bundles and muscle fibers of the dermis were wavy and well-organized. After 8 weeks of UV irradiation, the histologic features of epidermal and dermal thickening, associated with a dramatic loss of muscle fibers in the dermis, were detected, especially within the MC group. The dermal layer was poorly organized, and the sebaceous gland shrunk, which was consistent with previous reports [25,26]. In addition, inflammatory infiltrate ranged in composition from dermis to adipocytes, and hair follicles. Treatment with L-ascorbic acid and different doses of SPPs could reverse these histopathological changes to some degree. The histopathology of the SPP-L group remained acute photodamage; however, the onset of regeneration was evident. In the SPP-M and SPP-H groups, the repair of the epidermis and dermis was more remarkable. There was a considerable increase in the number of collagen bundles and muscle fibers, which were well-organized in the SPP-H group. The elastic fiber aggregation and hyperplasia caused by UV irradiation were conspicuously ameliorated. Consequently, the abnormal proliferation of the epidermis was effectively alleviated, and the dermis was well-protected [27]. A remarkable repair of the epidermis and dermis was observed after the application of SPPs, indicating that SPPs could effectively prevent UV-induced skin damage. Similar protective functions of peptides have been reported elsewhere [28,29].

### 3.5. Effect of SPPs on the Thymus and Spleen Index

The thymus and spleen are important in the immune system. T-lymphocytes developed in the thymus participate in monitoring and eliminating pathogenic microorganisms [30]. The spleen hosts a wide range of immunologic functions alongside its roles in hematopoiesis and red blood cell clearance [31]. As shown in Figure 3, both thymus and spleen index declined evidently in the MC group, indicating that UV-irradiation induced thymus and spleen damage. Supplementation with SPPs could improve the thymus and spleen index in a dose-dependent way, elucidating that SPPs could alleviate immunosuppression. Administration of protein hydrolysate might affect the thymus and spleen index or not; thus, we speculated that the differences are probably due to hydrolysis parameters and protein source [32,33].

### 3.6. ROS, MDA Levels, and Antioxidant Enzyme Activities

Long-term UV exposure will produce excessive amounts of reactive oxygen species (ROS) in skin tissues and lead to oxidative stress [34]. The increasing ROS levels can inhibit antioxidant defense mechanisms and cause oxidative damage associated with skin aging, liver damage, and even cancer [35]. The analyses of ROS and antioxidant enzymes are shown in Figure 4. The maximum contents of SOD and MDA were observed in the MC group, while L-ascorbic acid and SPPs reduced these indexes remarkably, as illustrated in Figure 4a,b. Notably, the high dosage of SPPs was as effective as L-ascorbic acid, indicating that SPPs could eliminate excessive production of ROS and suppress the increase in MDA caused by UA.

The level of ROS is strictly regulated by antioxidant enzymes such as superoxide dismutase (SOD), glutathione (GSH), and catalase (CAT), which maintain the intracellular environment [36]. Once the ROS level exceeds the ability of antioxidant cellular defense, oxidative stress occurs in the deeper layers of the skin. As estimated in Figure 4c–e, the SOD, GSH-Px, and CAT activities in the MC group showed a significant downward trend, indicating the existence of photooxidation caused by UV radiation. Compared with the MC group, the supplementation with SPPs and L-ascorbic acid improved the SOD, GSH-Px, and CAT activities to varying degrees, which means that SPPs can promote antioxidant enzyme activities to repair skin injury. As to the improvement in the SOD and CAT activities, the high dosage of SPPs obtained a similar effect compared to L-ascorbic acid. Meanwhile, even a low dosage of SPPs could effectively renovate the activities of SOD and GSH-Px, but the function in CAT was not sufficient.

### 3.7. Analysis of Levels of Inflammatory Cytokines

Skin inflammation induced by chronic UV exposure is one of the main factors associated with the progress of skin photoaging. Thus, the proinflammatory cytokine IL-6 and anti-inflammatory cytokine IL-10 in the skin tissues were investigated, as illustrated in Figure 5. Notably, the level of IL-6 in the MC group was almost four times higher than that of the NC group. Compared with the MC group, the levels of IL-6 in the SPP treatment groups were reduced obviously. The level of IL-10 in the MC group was significantly lower than in the NC group. Although the level of IL-10 declined to a certain extent after administration of SPP, there were no statistical differences existed between the MC group and the medium- and low-dosage groups.

### 3.8. Biomarkers Related to Collagen Stability and Degradation

Collagen is the major structural fibrous insoluble protein in the extracellular matrix of skin; therefore, a change in collagen content is one of the important indicators of skin condition. Hydroxyproline (Hyp) is a reliable biomarker of collagen synthesis and degradation [32], and thus its content was investigated. As shown in Figure 6a, the content of Hyp decreased from 25.00 µg/mL to 10.75 µg/mL after UV radiation, strongly suggesting collagen degradation. The administration of SPPs could improve the content of Hyp; however, there is no statistical difference between the MC group and the SPP-M and SPP-L groups. Nevertheless, high dosages of SPPs and L-ascorbic acid could conspicuously alleviate the reduction in Hyp content induced by UV radiation.

Despite Hyp, matrix metalloproteinase (MMP) levels are another resultful biomarker related to collagen synthesis and degradation. MMPs are zinc-containing endopeptidases that mediate the degradation of extracellular matrix proteins, among which MMP-1 is involved in collagen type I and III degradations, and MMP-9 is involved in the degradation of collagen type IV [37]. The increasing synthesis and expression of MMPs induced by UV irradiation is a warning signal of the impairment of the collagen-rich dermis. The MMP-1 and MMP-9 secretions are shown in Figure 6b,c. The level of MMP-1 and MMP-9 in MC groups strikingly improved 3.3 times and 5.5 times compared to the NC group, which is consistent with other reports [38], indicating the degradation of the dermal extracellular matrix, particularly types I and III. This result is in accordance with the pathological analysis, which hints at an impediment to collagen synthesis. L-ascorbic acid and SPP treatment could restore the MMP secretion to relatively normal levels. Inhibiting the aberrant expression of MMPs represents a significant approach to investigating the effects of anti-photoaging treatments on the skin [39].

## 4. Conclusions

Silkworm pupae are valuable edible insects. Silkworm pupae, regarded as by-products for a long period, are a good source of protein, lipids, and minerals. With the increasing demand for sustainable animal-derived dietary protein, silkworm pupae are potentially widely used in the food industry because of their high nutritional value and various biological activities.

The present study demonstrated that silkworm pupae protein hydrolysates resulted in a strong beneficial effect on alleviating and repairing skin photoaging. SPPs were composed of 900 peptides, and the molecular distribution ranged from 300 to 1600 Da. Macroscopic visual appearance certified that SPPs could alleviate the structural damage caused by photoaging. The contents of ROS and MDA, as well as antioxidant indicators, were all regulated to normal levels after administration of SPPs, manifesting that SPPs renovated skin photoaging by inhibiting ROS production. SPPs could regulate the amounts of Hyp and MMPs, which means SPPs had a certain protective effect on collagen stabilization. By calculating the thymus and spleen index of mice, it was observed that SPPs could alleviate the immunosuppressive effects induced by UV radiation. The histological results demonstrated that SPPs could reduce the degree of skin damage and prevent excessive increases in the skin epidermis and dermis thickness.

In conclusion, the peptides from silkworm pupae protein decreased the cytokines IL-6 and IL-10 associated with inflammation and increased the content of antioxidant enzymes, including SOD, GSH, and CAT. These multi-target mechanisms suggested that SPPs would be an effective anti-photoaging material. However, this study has a limitation: the active peptide sequences of SPPs were not identified and quantified. Further investigation is needed to determine the active peptides.

## Figures and Tables

**Figure 1 foods-13-01971-f001:**
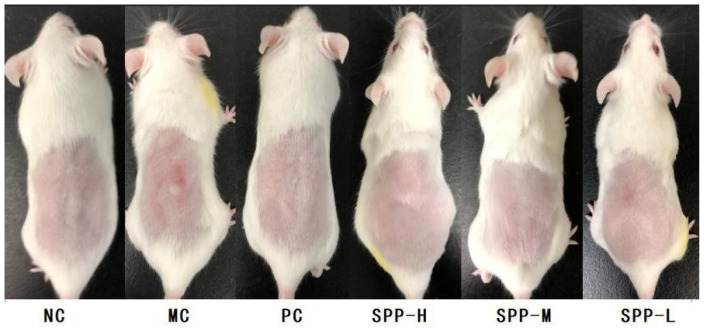
The macroscopic appearance of the dorsal skin of mice.

**Figure 2 foods-13-01971-f002:**
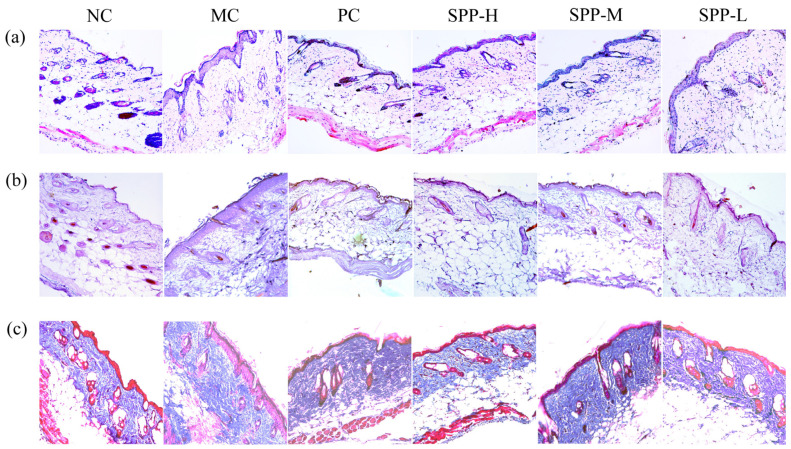
Histology of skin samples from photoaging mice (×100): (**a**) H&E staining; (**b**) Gomori staining; (**c**) Masson staining.

**Figure 3 foods-13-01971-f003:**
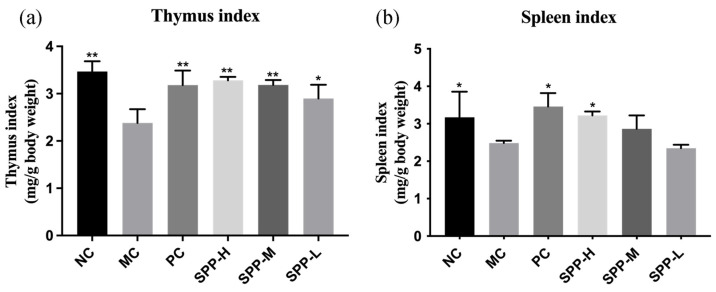
Effect of SPPs on the thymus and spleen index of photoaging mice. (**a**) Thymus index; (**b**) spleen index. Note: * *p* < 0.05, ** *p* < 0.01.

**Figure 4 foods-13-01971-f004:**
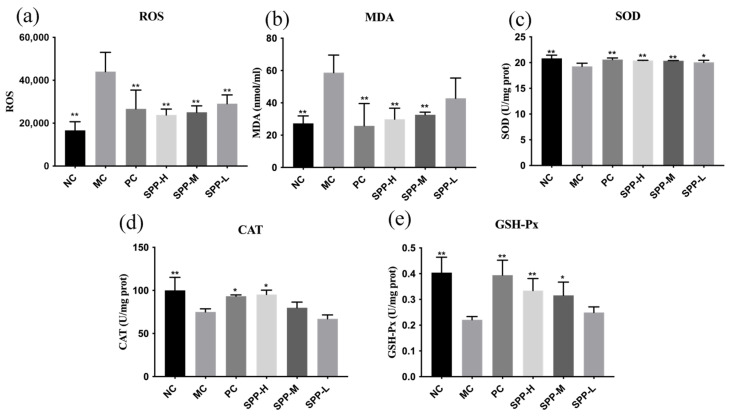
Effects of SPPs on serum ROS (**a**), MDA (**b**), SOD (**c**), CAT (**d**), and GSH-Px (**e**). Note: * *p* < 0.05, ** *p* < 0.01.

**Figure 5 foods-13-01971-f005:**
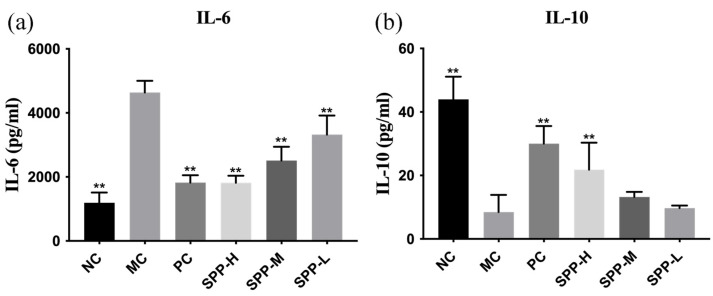
Effect of SPPs on the level of IL-6 (**a**) and IL-10 (**b**) in photoaging mice. Note: ** *p* < 0.01.

**Figure 6 foods-13-01971-f006:**
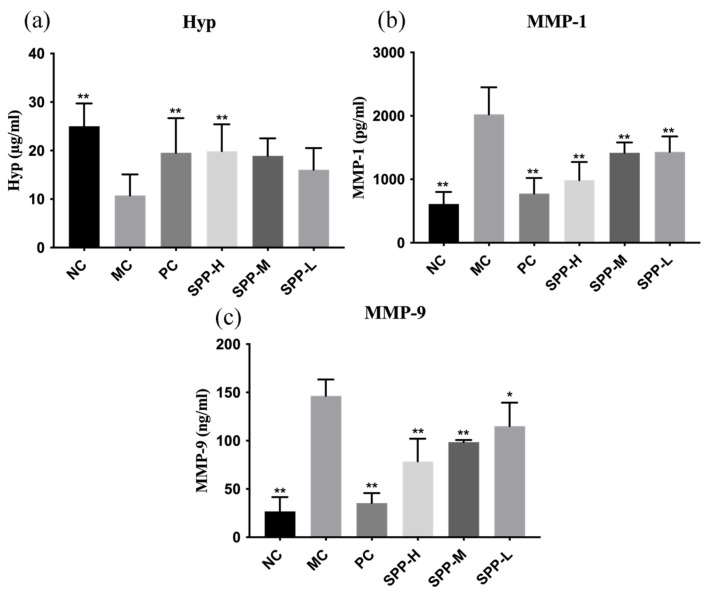
Effect of SPPs on the level of hydroxyproline (**a**), MMP-1 (**b**), and MMP-9 (**c**) in photoaging mice. Note: * *p* < 0.05, ** *p* < 0.01.

**Table 1 foods-13-01971-t001:** Experimental values and code levels of the independent variables.

Level	Parameters			
	A Temperature (°C)	B pH	C Enzyme Dosage (kU/g)	D Time (h)
−1	35	5	3	3
0	55	7	4	4
1	65	9	5	5

**Table 2 foods-13-01971-t002:** The grading scale for evaluation of photoaging.

Grade	Evaluation Criteria
0	Smoothness without any wrinkles; fine striations running the length of the body
1	Fine striations
2	A few shallow wrinkles; disappearance of all fine striations
3	Shallow wrinkles across the dorsal skin
4	Deep and coarse wrinkles with laxity
5	Increased deep wrinkles
6	Surface accompanied by severe wrinkles; development of lesions

**Table 3 foods-13-01971-t003:** The most abundant 10 peptides in SPPs.

Peptide	Mass	Length	*m*/*z*	RT	Area
LGKVYDY	856.433	7	429.2247	31.02	1,230,000,000
FLEP	504.2584	4	505.2655	29.69	857,000,000
WNEP	544.2281	4	545.2354	29.96	798,000,000
LYGDGDGSF	929.3766	9	930.3835	39.65	700,000,000
FLTPF	623.3318	5	624.3387	54.52	659,000,000
FVTPF	609.3162	5	610.3238	50.32	655,000,000
LPPF	472.2686	4	473.2761	43.75	640,000,000
LGPLLDPANER	1193.6404	11	597.828	37.28	627,000,000
LRLP	497.3326	4	249.6733	30.05	609,000,000
LRLP	559.3006	5	560.3079	36.29	560,000,000

**Table 4 foods-13-01971-t004:** The visual scores, pinch test, and skin thickness.

Group	Visual Score	Recovery Time (s)	Skin Thickness (nm)
NC	0.00 ± 0.00 ****	2.17 ± 0.20 ****	0.41 ± 0.02
MC	5.17 ± 0.69	7.05 ± 0.29	1.10 ± 0.16
PC	2.17 ± 0.37 ****	2.92 ± 0.25 ****	0.58 ± 0.05
SPP-H	3.17 ± 0.37 ****	4.27 ± 0.44 ****	0.68 ± 0.03
SPP-M	3.50 ± 0.50 ****	5.37 ± 0.37 ****	0.79 ± 0.03
SPP-L	4.33 ± 0.47 *	6.52 ± 0.49	0.87 ± 0.03

Note: * *p* < 0.05, **** *p* < 0.0001.

## Data Availability

The original contributions presented in the study are included in the article/Appendix A, further inquiries can be directed to the corresponding author.

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
