# Peer review of "The Protective Effects of Silkworm (Bombyx mori) Pupae Peptides on UV-Induced Skin Photoaging in Mice"

_foods, 2024, doi:10.3390/foods13131971_

Round 1
Reviewer 1 Report
Comments and Suggestions for Authors
· Interesting subject of the manuscript, in general some parts of the article are very clear, but in others I consider that it lacks quality. In the results and discussion of the manuscript this is essentially missing. Lack of scientific weight.
· Use the scientific name of the species (silkworm).
· In abstract line 13-14 it is not known what SPP means, it was not mentioned before.
· I suggest improving the objective at the end of the introduction
· The methodology described in line 140-143 does not seem objective to me, I suggest using another method.
· Line 296, methodology.
· Methodology cannot be included in the conclusion.
· The conclusion is generally of low quality, mixes a lot of issues and has little relationship with the objective of the investigation.
· Review the connection of each part of the manuscript, the title, objective, methodology and conclusion.
· The results and discussion focus more on the damage from UV radiation and not on the benefits of SPP, which is their work, which is their research.
· I CONSIDER THAT THERE IS STILL A LONG WAY TO GO TO DISCUSS THE REASONS FOR THE BENEFITS OF THE SPP.
· Figures 1 and 2 are mainly very illustrative, very clear. UV damage is clearly illustrated
Author Response
Dear reviewer:
Thank you so much. We have carefully revised and checked our manuscript and answered all questions point by point. We highlighted the changes. Please kindly refer to the manuscript and see Response to the Reviewers’ Comments for the details below. We greatly appreciate the reviewer’s thoughtful advice and comments to help improve our study. We would like to re-submit the manuscript for consideration of publication. We would assure that all of the authors have read and approved the final submitted manuscript.
Q1. Interesting subject of the manuscript, in general some parts of the article are very clear, but in others I consider that it lacks quality. In the results and discussion of the manuscript this is essentially missing. Lack of scientific weight.
Answer: Thank you very much for reviewing our research from point to point. The results and discussion have been polished. We also have checked the language and formatting to ensure the sentences are coherent.
Q2. Use the scientific name of the species (silkworm).
Answer: As suggested, the scientific name of the silkworm, Bombyx mori, has been added to the title (Line 2) , Abstract (Line 11), Introductin (Line 29).
Q3. In abstract line 13-14 it is not known what SPP means, it was not mentioned before.
Answer: As suggested, a more detailed explanation of SPP has been included in the abstract in Line 14.
Q4. I suggest improving the objective at the end of the introduction
Answer: As suggested, the objective at the end of the introduction was rewritten from Lines 73 to 78. The language and format have been polished to have a better flow.
Q5. The methodology described in line 140-143 does not seem objective to me, I suggest using another method.
Answer: We revised this section to make it easier to understand.
Q6. Line 296, methodology.
Answer: AS suggested, “The MMP-1 and MMP-9 secretions were measured by ELISA as shown in Figure 6b” has been revised to “The MMP-1 and MMP-9 secretions are shown in Figure 6b”. The methodology has been removed from Line 300.
Q7. Methodology cannot be included in the conclusion.
Answer: Thank you for your suggestion. All methodologies have been removed from the conclusion part as shown in Lines 310, 311, and 323. The conclusion has been restructured to make it more fluent.
Q8. The conclusion is generally of low quality, mixes a lot of issues and has little relationship with the objective of the investigation.
Answer: By the suggestion, the conclusion has been rewritten. The main results of the study are reserved, and the unnecessary parts have been removed as shown from Lines 310 to 331.
Q9. Review the connection of each part of the manuscript, the title, objective, methodology and conclusion.
Answer: Thank you for your suggestion. We have reviewed each part of the manuscript, including the title, objective, methods, results, and conclusion. The language and the formatting have been polished and highlighted in the manuscript.
Q10. The results and discussion focus more on the damage from UV radiation and not on the benefits of SPP, which is their work, which is their research.
Answer: Thank you for your suggestion. This work focuses on the anti-photoaging effects of SPP, and we double-checked the whole manuscript to ensure that the key points are coherent. The revisions are highlighted in the manuscript.
Q11. I CONSIDER THAT THERE IS STILL A LONG WAY TO GO TO DISCUSS THE REASONS FOR THE BENEFITS OF THE SPP.
Answer: Thank you for your question. Silkworms are biological and economical edible insects. Silkworm pupae, regarded as by-products for a long period, are a good source of protein, lipids, and minerals. With the increasing demand for sustainable animal-derived dietary protein, silkworm pupae are potentially widely used in the food industry because of their high nutritional value and various biological activities. As we know, protein hydrolysates produced by enzymatic hydrolysis by appropriate enzymes under controlled conditions would isolate desired and potent bioactive peptides. Therefore, SPP as hydrolysates of silkworm pupae protein, may have unique bioactive and therapeutic values for researchers to discover. We have enriched the conclusion part as suggested.
Q12. Figures 1 and 2 are mainly very illustrative, very clear. UV damage is clearly illustrated
Answer: Thank you for your comments.
Reviewer 2 Report
Comments and Suggestions for Authors
The main criticisms of your work are:
* This study is a simple exercise.
* The scarce discussion of the presented results.
Since the idea and information provided of this current paper titled “The biological effect of silkworm pupae peptides on UV-induced skin photoaging in mice» are interesting. But, some points which should be addressed in order to improve the quality of the MS
Abstract section
1. The abstract should be revised and be concise
2. Please avoid to use abbreviations
3. What is the main objective of this study?
4. Authors should stress the novelty of this work
5. Some technical approaches used in this investigation should be developed briefly
6. I invite authors to add some numerical data
7. A comparative data should be introduced
8. A 2-3 concise and conclusive sentences should be added at the end of the abstract
- Introduction section
9. This sections should be rewritten, it is very short and please check English language,
10. I invite authors to use recent and proper references (2019-2023), and more sentences should be developed
11. Authors should stress the novelty of this work since several studies developed the same idea
12. Authors should discuss the practical application of studied matrix
13. How about the limit of SPP?
14. Some recent studies describing as comparison of some extracts by SPP of the studied matrix should be introduced
15. The objective was not clear, improve it
Material and methods section
16. This part should be rewritten; some parts should be concise, it is better to simplify some sub sections.
17. table 1. Please explain the choice of such level of the 4 studied parameters
18. All parameters should be linked. I invite authors to use proper tools as PCA, HCA , Correlation Pearson or heat maps, …
Results and Discussion section
19. The discussion of the results is described very briefly without any thoughts or conclusions as to why this may be so. The results are only described in the form of what came out.
20. Compare means by ANOVA, of all tables and figures
21. all data should be linked, since this study has several experiments
22. authors should deeply discuss their results, and compare their results with another recent and suitable works
23. poor quality of some figures!
The conclusion part
24. should be improved taking into all remarks and suggestions
Author Response
Dear reviewer:
Thank you so much. We have carefully revised and checked our manuscript and answered all questions point by point. We highlighted the changes. Please kindly refer to the manuscript and see Response to the Reviewers’ Comments for the details. We greatly appreciate your thoughtful advice and comments to help improve our study. We would like to re-submit the manuscript for consideration of publication. We would assure that all of the authors have read and approved the final submitted manuscript
Abstract section
- The abstract should be revised and be concise
Answer: Thank you for your suggestion. Extra abbreviations have been removed, and we also checked the language and formatting to make the abstract more concise.
- Please avoid to use abbreviations
Answer: Thank you for your suggestion, some of the abbreviations were removed from the abstract part. We still keep the abbreviation of silkworm pupae peptides (SPP), and we explained the meaning in Line 15.
- What is the main objective of this study?
Answer: This study aimed to investigate the anti-photoaging properties and the potential mechanism of silkworm pupae peptides. We emphasized that in the abstract part as suggested, please see Lines 13 to 15.
- Authors should stress the novelty of this work
Answer: Thanks for the suggestion. Silkworm pupae protein is a new type of protein resource, and there is still plenty of unknown information about its functionalities and potential applications. In this work, we investigated one of its important functions, anti-photoaging, and demonstrated the potential mechanism. This work and the results will shed light on the further utilization of silkworm pupae protein in skin photoaging precaution and therapeutic. We emphasized the novelty in the abstract as shown in Line 24 and Line 25.
- Some technical approaches used in this investigation should be developed briefly
Answer: As suggested, the technical approaches in the abstract have been reorganized.
- I invite authors to add some numerical data
Answer: Thanks for the suggestion. The figures and tables in the manuscript can explain the results clearly. So, it is not necessary add more numerical data.
- A comparative data should be introduced
Answer: Some comparative date has been introduced in the main part of the revised article as suggested.
- A 2-3 concise and conclusive sentences should be added at the end of the abstract
Answer: As suggested, conclusive sentences have already been added to the end of the abstract as shown in Line 22 and Line 23.
- Introduction section
- This sections should be rewritten, it is very short and please check English language,
Answer: As suggested, the introduction has been rewritten. We checked the language and formatting. Some more recent research also has been added to this part to stress the novelty of our work.
- I invite authors to use recent and proper references (2019-2023), and more sentences should be developed
Answer: Thank you for the suggestion. The references before 2019 have been replaced with other more recent papers, and we also cited some more references. Please check references 3, 8, 11, 12, 13, 14, 15, 20, 25, 29, 30, 34, 35, 38.
- Authors should stress the novelty of this work since several studies developed the same idea
Answer: Thank you for the question. Researchers have found the silkworm pupae protein hydrolysate and peptides possess multiple biological functions, such as enhancing immunity, anti-tumor, blood pressure regulating, and blood sugar control. However, the anti-photoaging properties and mechanisms have not been confirmed. This work will provide a systematic analysis of the protective effects of silkworm pupae peptides on skin photoaging via in vivo study, which will be beneficial for the development of silkworm pupae protein in the food industry. We also stress the novelty in the introduction part.
- Authors should discuss the practical application of studied matrix
Answer: The potential application of this study has been added to the introduction part.
- How about the limit of SPP?
Answer: Silkworms are biological and economical edible insects, and they contain various nutrient substances, such as protein, lipids, and minerals. SPP is the mixture of bioactive peptides isolated from silkworm pupae protein hydrolysate. It has great potential in the application of anti-photoaging, however, it still has limitations. We did not purify and isolate single peptides. It would be better if the most active peptide could be isolated and synthesized. Though there are some limits of SPP, it is worth noticing that purification and isolation of SPP have already started in our group.
- Some recent studies describing as comparison of some extracts by SPP of the studied matrix should be introduced
Answer: Thank you for the suggestion, some recent studies about the extracts of silkworm pupae protein have been introduced and compared.
- The objective was not clear, improve it
Answer: As suggested, the objective in the introduction has been restructured to make it clearer.
Material and methods section
- This part should be rewritten; some parts should be concise, it is better to simplify some sub sections.
Answer: As suggested, the material and methods section has already been reorganized. We added the catalog number of commercial kits and removed some extra parts.
- table 1. Please explain the choice of such level of the 4 studied parameters
Answer: The 3 levels of the 4 variables were designed according to the optimal conditions for the Alcalase. From the single-factor experiments (not provided in the article) we did, the optimal hydrolysis conditions for Alcalse were 50℃, pH 7.5, and 4 h, thus we did orthogonal tests as Table 1 illustrates.
- All parameters should be linked. I invite authors to use proper tools as PCA, HCA , Correlation Pearson or heat maps, …
Answer: Thank you for your good suggestion. Because, the tools use as PCA, HCA , Correlation Pearson or heat maps might good for the a large data processing. But in this study, the data is not large, so we not used these tools.
Results and Discussion section
- The discussion of the results is described very briefly without any thoughts or conclusions as to why this may be so. The results are only described in the form of what came out.
Answer:Thanks for your suggestion. The results and discussion part have been rewritten. The results have been discussed deeply and compared with other similar research.
- Compare means by ANOVA, of all tables and figures
Answer:As suggested, all the compare means have already been added to the tables and figures. Thank you for the suggestion.
- all data should be linked, since this study has several experiments
Answer:Thank you for your good suggestion. The tools use as PCA, HCA , Correlation Pearson or heat maps might good for the a large data processing. But in this study, the data is not large, so we not used these tools to link the data.
- authors should deeply discuss their results, and compare their results with another recent and suitable works
Answer:Thanks for your suggestion. The results and discussion part have been rewritten. The results have been discussed deeply and compared with other similar research.
- poor quality of some figures!
Answer:Thank you for the suggestion. All the figures have been re-constructed.
The conclusion part
- should be improved taking into all remarks and suggestions
Answer:By the suggestion, the conclusion has been rewritten. The main results of the study are reserved, and the unnecessary parts have been removed as shown from Lines 315 to 336.
Reviewer 3 Report
Comments and Suggestions for Authors
What is the „biological effect on UV-induced skin photoaging? Photoprotective effects or anti-photoaging effects?
The first sentences in the Abstract and the Introduction are unclear.
At the end of the Introduction, the novelty of this study should be clarified, and the research hypothesis should be defined.
What was used as a positive control to evaluate the anti-tyrosinase activity?
Is there any data on feed characteristics during the experiment?
Add manufacturing data for commercial kits for oxidative stress and inflammatory markers
Subtitle 2.2. should be rephrased to reflect content. Generally, optimization of the enzymatic hydrolyses of silkworm pupae protein should be highlighted throughout the whole manuscript.
Lines 169-179: no need for explanation of statistical data
Figures and Tables: provide an explanation for abbreviations and statistical significance
Comments on the Quality of English LanguageThe manuscript could greatly benefit from professional English editing. Some sentences are understandable.
Author Response
Dear reviewer:
Thank you so much. We have carefully revised and checked our manuscript and answered all questions point by point. We highlighted the changes. Please kindly refer to the manuscript and see Response to the Reviewers’ Comments for the details below. We greatly appreciate the reviewer’s thoughtful advice and comments to help improve our study. We would like to re-submit the manuscript for consideration of publication. We would assure that all of the authors have read and approved the final submitted manuscript
Q1. What is the „biological effect on UV-induced skin photoaging? Photoprotective effects or anti-photoaging effects?
Answer: Thank you for your question. The biological effect means the anti-photoaging effect. The title has been modified to make it easier for comprehension.
Q2. The first sentences in the Abstract and the Introduction are unclear.
Answer: The first sentences in the Abstract (Line 11) and the Introduction (Line 29) have been modified. Thank you very much for taking the time to point out our mistakes.
Q3. At the end of the Introduction, the novelty of this study should be clarified, and the research hypothesis should be defined.
Answer: Thank you for your suggestion. The objective and hypothesis of this study have been clarified, and the introduction has been rewritten from Lines 73-78.
Q4. What was used as a positive control to evaluate the anti-tyrosinase activity?
Answer: Thank you for your good suggestion. In this study, we did not used a positive control to evaluate the anti-tyrosinase activity. Because we only to evaluated the anti-tyrosinase activity of SPP hydrolysates to obtain the optimum hydrolysis conditions. For the future study, set a positive control
Q5. Is there any data on feed characteristics during the experiment?
Answer: Thank you for your suggestion. Mice were fed with a mouse-only feed. It is not special. So we did not provide any data on feed characteristics.
Q6. Add manufacturing data for commercial kits for oxidative stress and inflammatory markers
Answer: Thank you for your suggestion. All catalog numbers and manufacturing data have been added to the commercial kits in the Materials part in Lines 87-90.
Q7. Subtitle 2.2. should be rephrased to reflect content. Generally, optimization of the enzymatic hydrolyses of silkworm pupae protein should be highlighted throughout the whole manuscript.
Answer: Subtitle 2.2 has been modified to “Extraction and preparation of SPP from silkworm pupae protein” to make it clearer, and we appreciate the comment.
Q8. Lines 169-179: no need for explanation of statistical data
Answer: The explanation of statistical data “Statistical significance was defined as p<0.05” has been removed from Line 168.
Q9. Figures and Tables: provide an explanation for abbreviations and statistical significance
Answer: The explanations for statistical significance have been added to the figures in Lines 247, 259, 283, and 307.
Round 2
Reviewer 2 Report
Comments and Suggestions for Authors
accept as it is